# Viral Co-Infections of Warthogs in Namibia with African Swine Fever Virus and Porcine Parvovirus 1

**DOI:** 10.3390/ani12131697

**Published:** 2022-06-30

**Authors:** Umberto Molini, Giovanni Franzo, Tirumala B. K. Settypalli, Maria Y. Hemberger, Siegfried Khaiseb, Giovanni Cattoli, William G. Dundon, Charles E. Lamien

**Affiliations:** 1School of Veterinary Medicine, Faculty of Health Sciences and Veterinary Medicine, University of Namibia, Neudamm Campus, Private Bag 13301, Windhoek 9000, Namibia; u.molini76@gmail.com (U.M.); mhemberger@unam.na (M.Y.H.); 2Central Veterinary Laboratory (CVL), 24 Goethe Street, Private Bag 18137, Windhoek 9000, Namibia; khaisebs@gmail.com; 3Department of Animal Medicine, Production and Health, University of Padova, Viale dell’Università 16, 35020 Legnaro, Italy; giovanni.franzo@unipd.it; 4Animal Production and Health Laboratory, Animal Production and Health Section, Joint FAO/IAEA Division, Department of Nuclear Sciences and Applications, International Atomic Energy Agency, P.O. Box 100, 1400 Vienna, Austria; t.b.k.settypalli@iaea.org (T.B.K.S.); g.cattoli@iaea.org (G.C.); c.lamien@iaea.org (C.E.L.)

**Keywords:** warthog, African swine fever, porcine parvovirus, co-infection, Namibia

## Abstract

**Simple Summary:**

Wild animals can transmit diseases to domestic animals. In Africa, warthogs are known to be carriers of pathogens that can infect pigs; consequently, it is important to identify these pathogens in order to protect pigs from infection. In this study, two important swine pathogens i.e., African swine fever virus (ASFV) and porcine parvovirus 1 (PPV1) were identified in warthogs in Namibia and characterized genetically. The results will be of interest to those working in swine disease management and control in Namibia.

**Abstract:**

Understanding virus circulation in wild animals, particularly those that have contact with domestic animals, is crucial for disease management and control. In Africa, warthogs are known to be asymptomatic carriers of porcine pathogens; a recent study in Namibia has shown them to be positive for Porcine circovirus-2 (PCV-2). In this study, the same samples used for the PCV-2 investigation in Namibia were further screened for the presence of African swine fever virus (ASFV) and porcine parvovirus 1 (PPV1) by PCR. Of the 42 animals tested, 2 (4.8%) and 13 (31%) were positive for AFSV and PPV1, respectively. The two AFSV were also co-infected with PPV1. Combing the results of this study with the results of the previous PCV-2 investigation, four warthogs were shown to be co-infected with both PPV1 and PCV-2. Sequence and phylogenetic analysis revealed that the AFSV belonged to genotype (Ib) but were from different serogroups. Unexpectedly, the ASFVs from the warthogs were genetically distinct to those observed in an outbreak in the same region of Namibia that occurred less than fifteen months prior to the sampling of the warthogs. In fact, a stronger genetic relationship was observed between the warthog viruses and historical Namibian and South African ASFVs identified in 1980, 2004 and 2008. For the PPV1s, the closest relative to the Namibian PPV1 were viruses identified in wild boar in Romania in 2011. This study confirms that warthogs are carriers of porcine pathogens and the data should encourage further studies on larger populations of wild and domestic swine to more fully understand the epidemiology and transmission of viral pathogens from these species.

## 1. Introduction

The common warthog (*Phacochoerus africanus)* is widespread in arid regions of sub-Saharan Africa although the exact population size is unknown, [1]; in South Africa, estimates of over 22,250 animals have been proposed [2]. Known to be an asymptomatic carrier of African swine fever virus (ASFV) [3], studies have shown that warthogs can be serologically positive for *Mycobacterium bovis*, Foot and Mouth Disease, Rift valley fever, influenza A and Porcine parvovirus 1 (PPV1) [4,5,6]. As pig production increases in Africa, identifying other potential pathogens in warthogs and understanding the role that this species may play in disease transmission to domestic swine is fundamental to the development of targeted disease-control strategies.

ASFV is the sole member of the family *Asfarviridae* and the only known DNA arbovirus. It has had an enormous impact on swine farming in several countries in Africa, Europe and Asia due to its clinical outcome and the restrictions applied to limit its spread, which have included bans on animal movements and trade [3]. Currently, the virus has been reported in 35 African countries [7,8] where it is maintained in a sylvatic cycle involving *Ornithodoros* soft ticks and asymptomatically infected warthogs and bush pigs. Although several molecular epidemiological reports have characterized ASFVs circulating in African countries in recent years [9,10,11], only a few studies have focused on ASFV in warthogs [12,13,14].

Porcine parvovirus 1 (PPV1) is a small single-stranded non-enveloped DNA virus belonging to the genus *Parvovirus*, family *Parvoviridae*. The viral genome of approximately 5 Kb encodes three non-structural (NS1, NS2 and SAT) and three structural (VP1, VP2 and VP3) proteins. PPV1 is a major cause of reproductive disorders in pigs globally and has been shown to be responsible for embryonic death, stillbirth, mummification and infertility [15]. Although PPV1 has been found regularly in wild boar in Europe [16,17,18] and Asia [19,20] information of its presence in warthogs in Africa is limited [4].

This study was performed on a population of resident warthogs from two livestock/hunting farms in Namibia that had previously been shown to be infected with Porcine circovirus-2 (PCV-2) [21], which is a DNA virus that is associated with an assortment of different disease conditions, including PCV-2-reproductive disease, porcine dermatitis and nephropathy syndrome, and PCV-2-subclinical infection and PCV-2-systemic disease [22]. The objective of the study was to determine whether the same warthogs were co-infected with ASFV and PPV1.

## 2. Materials and Methods

### 2.1. Sample Description and Processing

This study included samples from 42 warthogs (*Phacochoerus africanus*) hunted between June and October 2019 from two contiguous livestock farms in Windhoek, Khomas region (Note: both of these farms experienced an ASF outbreak in domestic pigs at the beginning of 2018) [11]). Tonsils were collected during the slaughter phase at the farms’ abattoir and sent refrigerated to the Central Veterinary Laboratory of Windhoek (CVL) for further ASF investigation.

Samples (5 g of tonsil) were homogenized in 400 μL of sterile phosphate buffer saline (PBS) using a TissueLyser LT (Qiagen, Hilden, Germany). Total genomic DNA was extracted from the homogenized samples using a High Pure Viral Nucleic Acid Kit (Hoffman-La Roche, Basel, Switzerland) with an elution volume of 100 μL, following the manufacturer’s instructions.

### 2.2. PCR Screening

The presence of ASF-specific DNA was tested using a method adapted from King et al., [23], while a 739 bp fragment of the VP2 gene was amplified in PPV1 positive samples with primers PPV1-VP2-F1 5′ GGGAGGGCTTGGTTAGAAT 3′ and PPV1-VP2-R1 5′ CTGGTAGTGTTCCTGGGTGT 3′. The PCR reaction conditions for the PPV1 were: 5 μL of extracted DNA in a total reaction volume of 20 μL containing a final concentration of 1.25 mM MgCl_2_, 1X PCR buffer (Qiagen, Germany), 0.2 mM dNTPs, 10 pmol of each primer, and 2.5 U of Taq DNA polymerase. The reactions were performed with the following thermal profile: initial denaturation at 94 °C for 10 min, then 35 cycles of denaturation at 95 °C for 30 s, annealing at 54 °C for 30 s and elongation at 72 °C for 60 s, followed by a final elongation at 72 °C for 5 min

Positive ASFV samples were further amplified using four different primer pairs targeting: the C-terminal region of the B646L gene encoding the p72 protein [24], the full-length E183L (p54) gene encoding the p54 protein [25], the central variable region (CVR) of the B602L gene [26] and the partial CD2v gene [27].

### 2.3. Sequencing and Phlyogenetic Analysis

Amplicons were purified using the Wizard^®^ SV Gel and PCR Clean-Up System (Promega) and sequenced commercially by LGC Genomics (Berlin, Germany). All sequences generated in this study have been submitted to GenBank under accession numbers (OM176558-OM176565 and ON383314-ON383321). For the ASFV positive samples, the partial B646L (p72) and the full E183L (p54) gene sequences were used to determine the genotype of the viruses and evaluate potential epidemiological links by phylogenetic analysis and comparison with reference strains, while the CD2v sequence fragment was analyzed to determine the serogroup. All sequence alignments and phylogenetic analyses were performed using MEGA 7.

The neighbor-joining (NJ) tree for the B646L (p72) nucleotide sequences was constructed by selecting the Maximum Composite Likelihood (MCL) as the best substitution model, while the Kimura 2-parameter was preferred for the p54 alignment. For the CD2v gene, a maximum-likelihood (ML) tree of the partial amino-acid sequences was constructed applying the CpREV + G substitution model. The tetrameric tandem repeat sequences (TRS) within the CVR were extracted from the deduced amino-acid sequences of the partial B602L gene. Each TRS was transformed into a single letter code utilizing previously published codes for comparison [28,29]. For the VP2 sequence of the positive PPV1 samples, a NJ phylogenetic tree based on raw genetic distances (pairwise p-distance) was generated. The reliability of the sequence cluster was evaluated by performing 1000 bootstrap replicates.

## 3. Results and Discussion

The presence of ASFV and PPV1 DNA was confirmed in two (4.76%) and thirteen (35.7%) samples, respectively. The two ASFV positive samples (W6 and W17) were also positive for PPV1. Both samples were collected in one of the two farms where ASF outbreaks occurred in domestic pigs between February and March 2018 [11]. The phylogenetic tree of the C-terminal of the p72 gene revealed that Namibian warthogs ASFV belong to genotype I (Figure 1).

Interestingly, and unexpectedly, the analysis showed a stronger clustering of the warthog ASFV sequences with sequences from historical samples collected in Namibia and South Africa since 1980 from domestic pigs and warthogs than to the ASFVs identified in pigs in Namibia in 2018.

This unexpected finding was further confirmed using the p54 gene tree, which showed that the Namibian samples clustered together with an ASFV sampled in South Africa in 1985 (GenBank KJ671541) (genetic distance = 0.2%) (Figure 2).

Nevertheless, there was a genetic distance of only 0.8% with the ASFV identified in pigs during the 2018 Namibian outbreak. The persistent circulation and replication in warthogs for at least fifteen months (i.e., March 2018 to June 2019) between the domestic pig outbreak and the sampling of the warthogs could explain the low, but not negligible, genetic distance between strains identified in the warthogs, so it is not inconceivable that they could have played a role in introducing ASFV into the domestic pigs in 2018.

The warthog sequences belonged to different unassigned serogroups based on the analysis of the CD2v sequence (Appendix A). The two strains showed a different tetrameric tandem repeat profile (i.e., W6: BNAAAFBTDBNAFNBTFNBNAAAF and W17: BNAAAAAADBNAFNBNAAFA). Strain W6 showed the same profile as the Nam_2018_4593 and Nam_2018_5538, detected in domestic pigs during the previous clinical outbreak (Appendix A), while the profile for W17 was unique.

Partial VP2 sequence was generated for eight of the 13 PPV1 positive samples. Phylogenetic analysis of the sequences revealed that the Namibian viruses grouped with PPV1s identified in wild boar in Romania in 2011 and were genetically distinct to PPV1s identified in domestic pigs in Nigeria (the only other African country from which comparable PPV1 sequence data is available) (Figure 3). As there is no clear epidemiological link between Romanian wild boar and Namibian warthogs, a more in-depth genomic analysis of a larger sample group which should include samples from neighboring countries is required to clarify the true similarities between these viruses. Interestingly, the Namibian and Romanian PPV1 appear to be specific for wild swine, another observation that requires further investigation. At this point, it would also be opportune to screen domestic pigs in Namibia for PPV1 to determine if there is an epidemiological link between them and warthog PPV1s.

By combining the previous data on the presence of PCV-2 in warthogs [21], four were shown to be co-infected with both PPV1 and PCV-2c (Table 1). Co-infections of PPV1 and PCV-2 are commonly reported in domestic pigs [30]; such co-infections are evidently occurring in warthogs too. It is known that PPV-1 can aggravate infections of domestic swine with PCV-2 [31] but this does not appear to be the case in warthogs, as all of the animals appeared healthy prior to slaughter.

The control of ASF outbreaks in Namibia relies on strict compartmentalization (farm biosecurity), by separating domestic pigs from warthogs, and the stamping out of pigs in infected farms [32] but there is no official control strategy for PPV1. A vaccine (Porcilis^®^ Parvo) for PPV1 is registered by MSD Animal Health, Kempton Park, South Africa in Namibia but it has not yet been used by pig producers. Indeed, there is no data on PPV1 infections of domestic pigs in the country. It would be desirable if the current data encourage veterinarians and pig producers to investigate the presence of PPV1 in their stock and assess the risk of transmission from infected wildlife.

## 4. Conclusions

Warthogs have been confirmed as asymptomatic carriers of ASFV and PPV1. Although our findings have been unable to identify a strong epidemiological link between the ASFV positive warthogs and previous ASF outbreaks in pigs, warthogs should still be considered a risk for the introduction of viral pathogens to domestic swine.

## Figures and Tables

**Figure 1 animals-12-01697-f001:**
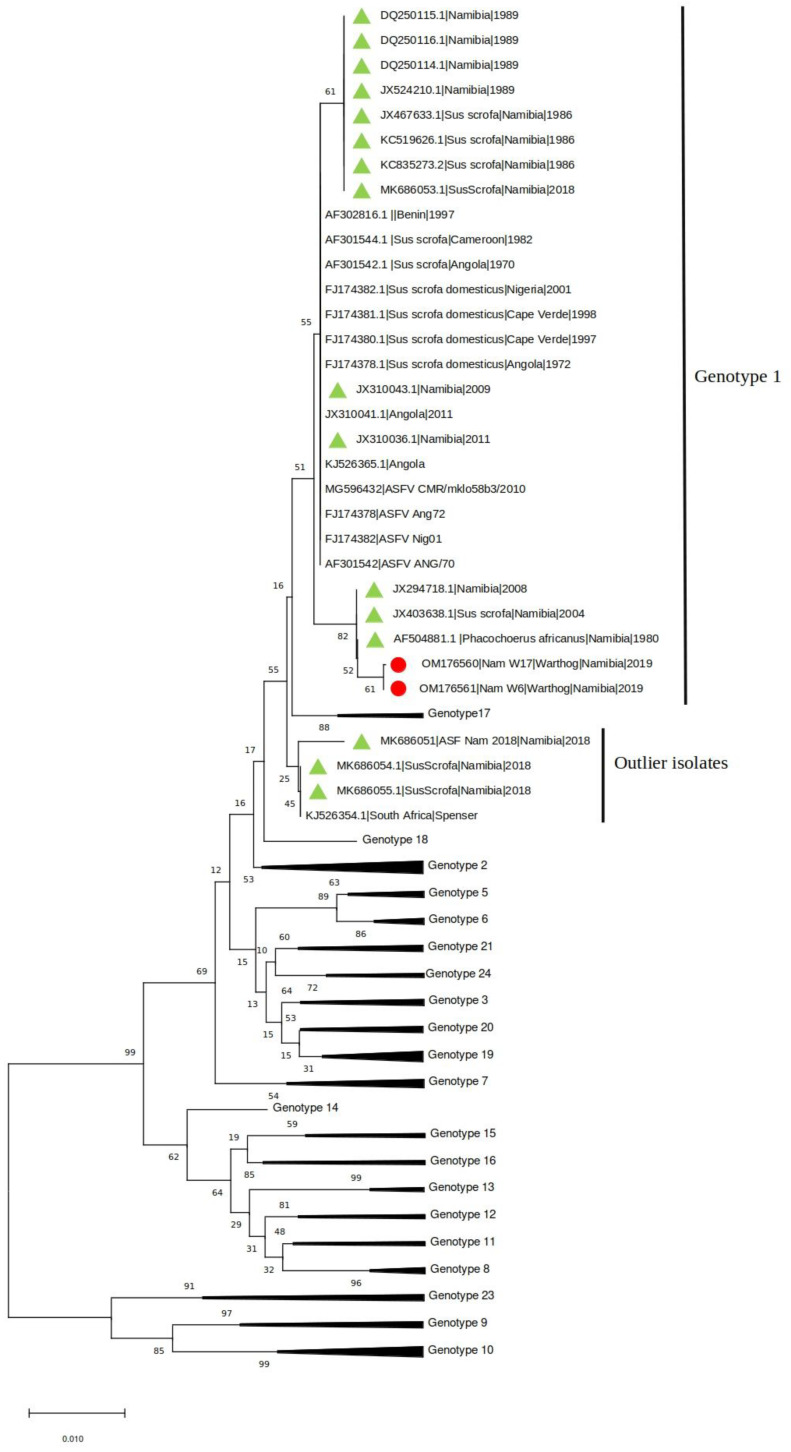
Neighbor-joining phylogenetic tree reconstructed based on the p72 sequence dataset. The sequences obtained in the present study have been highlighted with a red dot, while other Namibian samples are shown with a green triangle.

**Figure 2 animals-12-01697-f002:**
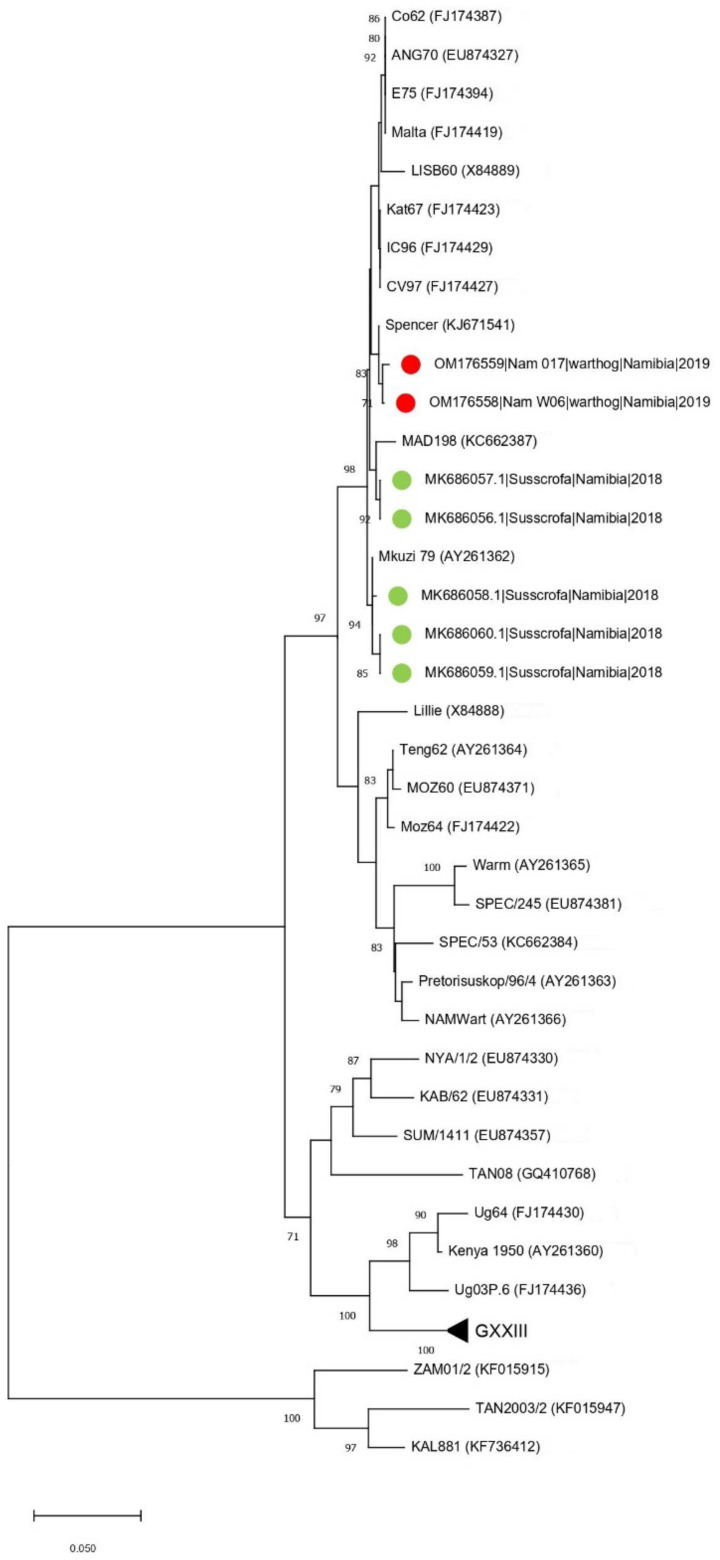
Neighbor-joining phylogenetic tree based on the p54 sequence dataset. The sequences obtained in the present study have been highlighted with a red dot, while other Namibian samples are shown with a green triangle.

**Figure 3 animals-12-01697-f003:**
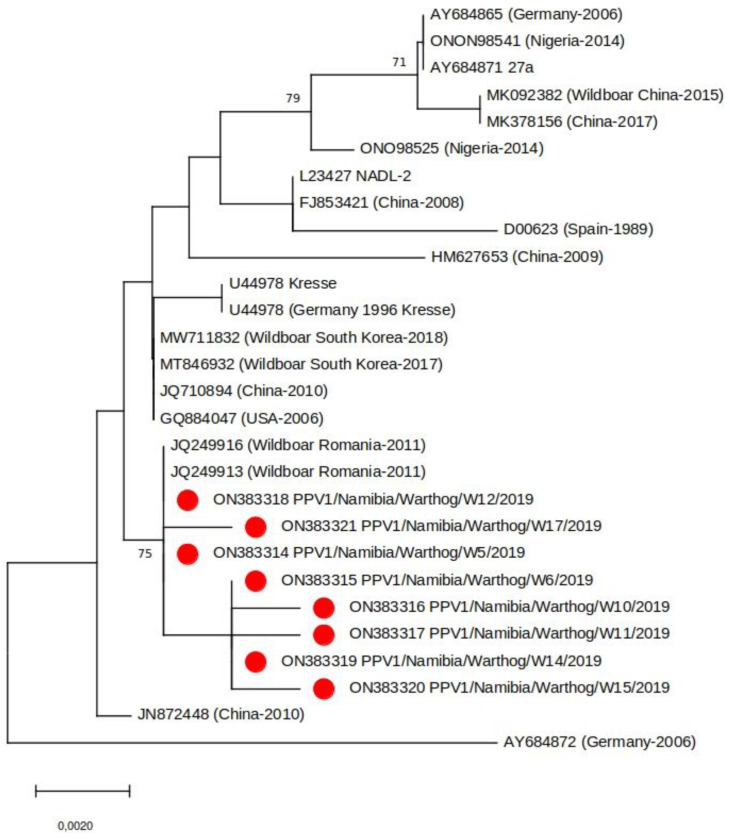
Neighbor-joining phylogenetic tree based on a subset of representative PPV1 VP2 sequences (683 bp). The samples from this study are indicated with a red circle. Bootstrap values (>70%) are shown.

**Table 1 animals-12-01697-t001:** Results of sample screening.

Sample	ASFV	PCV-2c *	PPV1	Sample	ASF	PCV-2c *	PPV1
W1	−	−		W22	−	+	−
W2	−	−	+	W23	−	+	−
W3	−	−	+	W24	−	−	−
W4	−	−	+	W25	−	−	−
W5	−	−	+	W26	−	−	−
W6	+	−	+	W27	−	−	−
W7	−	−	−	W28	−	−	−
W8	−	+	−	W29	−	−	−
W9	−	−	−	W30	−	−	−
W10	−	−	+	W31	−	−	−
W11	−	+	+	W32	−	−	−
W12	−	+	+	W33	−	−	+
W13	−	−	−	W34	−	−	+
W14	−	+	+	W35	−	−	−
W15	−	+	+	W36	−	−	−
W16	−	+		W37	−	−	−
W17	+	−	+	W38	−	−	−
W18	−	−	−	W39	−	−	−
W19	−	−	−	W40	−	−	−
W20	−	+	−	W41	−	−	−
W21	−	+	−	W42	−	−	−

* The PCV-2c results are from the previous study from Molini et al. [21].

## Data Availability

Sequences generated in this study have been submitted to GenBank under accession numbers OM176558-OM176565 and ON383314-ON383321.

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
