# Peer review of "Viral Co-Infections of Warthogs in Namibia with African Swine Fever Virus and Porcine Parvovirus 1"

_animals, 2022, doi:10.3390/ani12131697_

Round 1
Reviewer 1 Report
The manuscript is well written and should be accepted after minor corrections. The content is useful to inform the epidemiology of ASF in Africa and elsewhere. Comments are inserted in the attached document.

Author Response
- Line 51: the text has been revised according to the reviewer’s comment
- Line 69: more information on the warthog population has been added as requested.
Reviewer 2 Report
This study investigated the Viral co-infections of warthogs in Namibia in 2019, and found that some warthogs were infected/co-infected by the ASFV and PPV1. Interestingly, the identified ASFV viruses were most similar to those isolated more than 10 years ago, instead of the ASFVs which caused the outbreak in 2018. This suggested the persistent circulations of ASFVs in warthogs or pigs. Besides, the PPV1 viruses were also more similar to those isolated in Europe instead of those in African countries, suggesting possible epidemiological links between Europe and African countries, or large gap in sampling of the virus. Overall, the study is valuable for understanding the circulation and evolution of both ASFVs and PPV1 in warthogs. The reviewer only has one minor concerns: L70, “;” should be “,”
Author Response

(The authors gave the same response as above.)

Reviewer 3 Report
Lines 69-70 : repeating "been", please removing unnecessary 'been'
Line 83: please provide the exact amount of the tonsils sample used for homogenisation and/or % dilution in PBS (w/v) of tonsils used for extraction
Lines 165-166: Rewrite the sentence "By combining the previous data on the presence of PCV-2 in warthogs, four were shown to be co-infected with both PPV1 and PCV-2c (Table 1)." In Table 1. there is no data of PCV-2c , in addition references regarding PCV-2 is missing in line 166. It is recommended to add informations in Table 1 (* footnote below the table 1) that PCV-2 results origin from previous study in which the same diagnostic material (tonsils) was used.
Lines 187-192: In my opinion, information about pcv2 is redundant as it is not the target of research.
Author Response
- Lines 69-70 : repeating"been", please removing unnecessary 'been'
Response: revised as requested
- Line 83: please provide the exact amount of the tonsils sample used for homogenisation and/or % dilution in PBS (w/v) of tonsils used for extraction
Response: the information has been added as requested
- Lines 165-166: Rewrite the sentence "By combining the previous data on the presence of PCV-2 in warthogs, four were shown to be co-infected with both PPV1 and PCV-2c (Table 1)." In Table 1. there is no data of PCV-2c , in addition references regarding PCV-2 is missing in line 166. It is recommended to add information in Table 1 (* footnote below the table 1) that PCV-2 results origin from previous study in which the same diagnostic material (tonsils) was used.
Response: The text and Table 1 have been revised as suggested
- Lines 187-192: In my opinion, information about pcv2 is redundant as it is not the target of research.
Response: The text has been revised taking into account the reviewer’s comment.